# Nifedipine Modulates Renal Lipogenesis via the AMPK-SREBP Transcriptional Pathway

**DOI:** 10.3390/ijms20071570

**Published:** 2019-03-29

**Authors:** Yen-Chung Lin, Mai-Szu Wu, Yuh-Feng Lin, Chang-Rong Chen, Chang-Yu Chen, Chang-Jui Chen, Che-Chou Shen, Kuan-Chou Chen, Chiung-Chi Peng

**Affiliations:** 1Graduate Institute of Clinical Medicine, College of Medicine, Taipei Medical University, Taipei 110, Taiwan; yclin0229@tmu.edu.tw (Y.-C.L.); maiszuwu@gmail.com (M.-S.W.); linyf@s.tmu.edu.tw (Y.-F.L.); 2Division of Nephrology, Department of Internal Medicine, Taipei Medical University Hospital, Taipei 110, Taiwan; 3Division of Nephrology, Department of Internal Medicine, School of Medicine, College of Medicine, Taipei Medical University, Taipei 110, Taiwan; 4Division of Nephrology, Department of Internal Medicine, Taipei Medical University-Shuang Ho Hospital, New Taipei City 235, Taiwan; 5International Medical Doctor Program, Vita-Salute San Raffaele University, 20132 Milan, Italy; cherylcherylchen@gmail.com; 6Wayland Academy, 101 North University Avenue, Beaver Dam, WI 53916, USA; cychen@wayland.org (C.-Y.C.); rayraycjc@gmail.com (C.-J.C.); 7Department of Electrical Engineering, National Taiwan University of Science and Technology, Taipei 106, Taiwan; choushen@mail.ntust.edu.tw; 8Department of Urology, Taipei Medical University-Shuang Ho Hospital, New Taipei City 235, Taiwan; 9Department of Urology, School of Medicine, College of Medicine, Taipei Medical University, Taipei 110, Taiwan

**Keywords:** nifedipine, renal lipotoxicity, AMPK, calcium channel blockers, lipin-1, sterol regulatory element-binding proteins 1/2 (SREBP1/2)

## Abstract

Lipid accumulation in renal cells has been implicated in the pathogenesis of obesity-related kidney disease, and lipotoxicity in the kidney can be a surrogate marker for renal failure or renal fibrosis. Fatty acid oxidation provides energy to renal tubular cells. Ca^2+^ is required for mitochondrial ATP production and to decrease reactive oxygen species (ROS). However, how nifedipine (a calcium channel blocker) affects lipogenesis is unknown. We utilized rat NRK52E cells pre-treated with varying concentrations of nifedipine to examine the activity of lipogenesis enzymes and lipotoxicity. A positive control exposed to oleic acid was used for comparison. Nifedipine was found to activate acetyl Coenzyme A (CoA) synthetase, acetyl CoA carboxylase, long chain fatty acyl CoA elongase, ATP-citrate lyase, and 3-hydroxy-3-methyl-glutaryl-coenzyme A (HMG CoA) reductase, suggesting elevated production of cholesterol and phospholipids. Nifedipine exposure induced a vast accumulation of cytosolic free fatty acids (FFA) and stimulated the production of reactive oxygen species, upregulated CD36 and KIM-1 (kidney injury molecule-1) expression, inhibited p-AMPK activity, and triggered the expression of SREBP-1/2 and lipin-1, underscoring the potential of nifedipine to induce lipotoxicity with renal damage. To our knowledge, this is the first report demonstrating nifedipine-induced lipid accumulation in the kidney.

## 1. Introduction

Ectopic lipid deposition in non-adipose tissues such as the kidney may elicit organ damage with subsequent lipotoxicity, further affecting the function of the involved organs [1]. Overproduction of free fatty acids (FFA) may enhance the translocation of FFA into mitochondria, triggering the production of reactive oxygen species (ROS) in the early phase of accumulation. This results in increased mitochondrial proton conductance and uncoupling of oxidative phosphorylation, eventually provoking mitochondrial permeabilization [2], which underlies dysfunction in fatty acid oxidation (FAO). Inhibition of FAO in renal proximal tubule epithelial cells causes depletion of ATP, cell death, de-differentiation, and intracellular lipid deposition, all of which contribute to the phenotype of renal fibrosis [3].

When kidney cells are injured, individual cell types are differently affected. The endothelial cells undergo apoptosis, inducing inflammation and mesangial cell proliferation and eventual glomerulopathy. Furthermore, podocyte apoptosis and ER stress may be evoked, resulting in proteinuria and glomerulopathy. Lastly, the tubular cells also undergo apoptosis with increased autophagy vesicles, leading to interstitial inflammation and fibrosis [4].

CD36 (cluster of differentiation 36, also known as platelet glycoprotein 4, fatty acid translocase (FAT) or scavenger receptor B2) is a scavenger receptor that functions in high-affinity tissue uptake of long-chain fatty acids (LCFAs) and, under conditions of excessive fat supply, contributes to lipid accumulation and metabolic dysfunction [5]. Renal CD36 is mainly expressed in the proximal tubule cells, podocytes, and mesangial cells, and is markedly upregulated in the setting of chronic kidney disease (CKD) [6]. Recently, growth hormone releasing peptides (GHRP) have been identified as potent inducers of PPAR-γ and its downstream actions through the activation of CD36, demonstrating an alternative method of regulating essential aspects of hepatic cholesterol biosynthesis and fat mitochondrial biogenesis [7].

Hepatic lipogenesis has been reported to be mediated predominantly by the AMP-activated protein kinase/sterol regulatory element-binding protein-1 (AMPK/SREBP-1) pathway in rat hepatocytes and human hepatoma cell lines [8]. A diet incorporating oleic acid (OA) decreased the phosphorylation of AMPK and increased the maturation of SREBP-1 and the expression of SREBP-responsive genes [8]. Mice and mouse hepatocytes were resistant to OA-induced lipogenesis because of little, if any, response by AMPK and downstream effectors [8].

More and more studies have linked the activation of sterol regulatory element-binding proteins (SREBPs), the dysfunction of peroxisome proliferator-activated receptors (PPARs), and inhibited β-oxidation of fatty acids with diabetic nephropathy [9]. High glucose may initiate lipid deposition on renal tubular cells by upregulating CD36 and PPARs [10]. SREBP-1c, regulated by the insulin and AMPK signaling pathways, is reported to play a role in nonalcoholic fatty liver disease [11]. In addition, lipin plays an important role in lipid metabolic homeostasis [12]. Lipin-1 has been shown to act as a key mediator of the effects of mTORC1 on SREBP-dependent fatty acid and cholesterol biosynthesis [13].

It has been well documented that many thiazide-type diuretic antihypertensives exhibit a hyperlipidemic effect, while such adverse effects are seldom associated with the calcium channel blockers (CCBs) [14,15]. Research suggests that CCBs may impair renal autoregulation [16], and clinically, they may be less effective than other antihypertensives [17]. Moreover, the safety of CCBs in patients with proteinuria and renal insufficiency is widely questioned [18]. We hypothesize that nifedipine may promote renal damage due to its lipogenic effect, which is associated with the AMPK-SREBP-lipin 1 pathway. In this study, using the normal rat proximal tubular epithelial cell line NRK52E as the model, we investigated whether the lipogenesis of nifedipine is associated with the AMPK-SREBP-lipin 1 pathway.

## 2. Results

### 2.1. Effect of Nifedipine on Cell Viability

Nifedipine suppressed the viability of NRK52E cells (Figure 1a). After incubation for 24 h and 48 h, nifedipine at low dose 7.5 μM did not show any effect on the viability of NRK52E cells. At 15 μM and 30 μM, nifedipine suppressed cell viability to 15.6% and 39.0%, respectively, relative to the control and to 40.3% of that induced by oleic acid (*p* < 0.05) (Figure 1a) during the first 24 h. The cell viability was further suppressed to 24.4% and 47.2% relative to the control after 48 h, compared with 42.3% caused by oleic acid (*p* < 0.05) (Figure 1a). Initially, there were higher cell viabilities at 48 h (up to 147.2%) than at 24 h for control, and then nifedipine 30 µM significantly decreased cell viabilities (62.5% in 24 h and 60.0% in 48 h) compared to control (*p* < 0.01). Nifedipine decreased cell viabilities in MTT assays.

### 2.2. Effect of Nifedipine on Intracellular Lipid Accumulation

Oil Red O staining revealed that the intracellular lipid content significantly increased in dose of 30 µM nifedipine and 150 µM Oleic acid in both 24 and 48 h groups (*p* < 0.05) (Figure 1b). However, similar quantifications of Oil Red O were shown in the low doses of 7.5 and 15 µM groups compared with untreated control (Figure 1b,c).

### 2.3. Nifedipine Stimulated Production of Reactive Oxygen Species (ROS)

Nifedipine at 30 μM stimulated the production of ROS at a 3.3-fold rate compared to the control group in ROS(+)/ROS(−) after treatment with nifedipine for 16 h, whereas a 2.57-fold increase was observed after treatment with 500 μM H_2_O_2_ as positive control for 30 min (*p* < 0.01) (Figure 2a).

### 2.4. Nifedipine Promoted the Expression of Proteins Related to Kidney Injury

The expression of KIM (kidney injury molecule-1; also known as T cell immunoglobulin mucin domains–1 (TIM-1)) was simultaneously upregulated to 101%, 102%, and 122% (*p* < 0.01) compared with the control, respectively, following nifedipine exposure at 7.5, 15, and 30 μM for 24 h, compared with 103% induced by oleic acid-positive control (Figure 2b) and then decreased to 86.0%, 91.0%, and 80% at 48 h (Figure 2c) relative to the control.

### 2.5. Nifedipine Stimulated Cholesterol De Novo Biosynthesis

Nifedipine stimulated de novo cholesterol biosynthesis in a non-dose- and non-time-dependent manner. The cholesterol level reached a range from 111.3 μg/mL (nifedipine 7.5 μM) to 114.9 μg/mL (nifedipine 30 μM), compared with 93.5–97.4 μg/mL for the control in 24 h or 48 h, respectively (*p* < 0.01) (Figure 3a).

### 2.6. Nifedipine Upregulated the Expression of CD36 Proteins Related to Lipid Translocation

CD36 was upregulated by nifedipine at 30 μM to 143.6% (*p* < 0.05), but unaffected at the lower doses 7.5 and 15 μM (Figure 3b). In addition to acting as a fatty acid translocase, CD36 also functions as a novel mediator influencing binding and uptake of albumin in the proximal tubule, and CD36 is upregulated in proteinuric renal diseases. Thus, the upregulation of CD36 induced by nifedipine (30 μM) in the rat kidney NRK52E cell line supports the hypothesis that nifedipine at regular therapeutic dosages could provoke renal cholesterol accumulation (Figure 3a,b).

To understand whether the cholesterol accumulating effect of nifedipine was due to stimulation of the de novo biosynthetic pathway or merely because of translocation, we further explored the effect of nifedipine on the de novo biosynthetic pathway of cholesterol.

### 2.7. Nifedipine Stimulated the Production of Acetyl CoA Synthetase (ACS), Phosphorylated Acetyl-CoA Carboxylase (p-ACC) over Total Acetyl-CoA Carboxylase Acetyl (ACC), and Long Chain Fatty Acyl Elongase (ACSL1), but Not Fatty Acid Synthase (FAS) in 48 h

Nifedipine stimulated the production of ACS (*p* < 0.05 at 7.5 μM and *p* < 0.01 at 15 μM and 30 μM) (Figure 4a) in 48 h. The ACC increased to 128.5% and 144.4% at doses 15 and 30 μM, compared to the control (*p* < 0.01), and p-ACC/ACC decreased to 81.5% and 74.1% at doses 15 and 30 μM, to the control because of inhibitory phosphorylation of ACC (*p* < 0.01) (Figure 4b). In addition, ACSL1 was expressed at 116.7%, 120%, and 126.7% at doses 7.5, 15, and 30 μM respectively (*p* < 0.01 at all concentrations) (Figure 4c), and depressed fatty acid synthase (FAS) at 83.7%, 79.1%, and 81.4% at doses 7.5, 15, and 30 μM, respectively (*p* < 0.05) (Figure 4d).

### 2.8. Nifedipine Stimulated the Production of ATP Citrate Lyase (ACL) and HMG-CoA Reductase (HMGCR)

The amount of ATP citrate lyase produced was stimulated by nifedipine in a dose dependent manner, with 7.5 and 15 μM doses increasing production to 150% and 169%, and then leveling off at 30 μM at 170%, compared with the control (*p* < 0.01) (Figure 5a). In parallel, the amount of downstream HMG-CoA reductase was also upregulated by nifedipine administration in a dose-dependent fashion to 107% and 120% at 15 and 30 μM doses as compared with the control (*p* < 0.01) (Figure 5b). This then led to increase of de novo cholesterol biosynthesis (Figure 3a,b).

### 2.9. Nifedipine Downregulated the Expression of Phosphorylated AMPK

Although the level of 5′ AMP-activated protein kinase (AMPK)/actin was slightly yet significantly upregulated in a dose-dependent fashion following treatment with nifedipine to 107.1%, 112.6%, and 116.4% of control levels, more significantly, the ratio of p-AMPK to AMPK was downregulated in a dose-dependent manner to 82.1%, 58.7%, and 49.9% of control levels, respectively, at 7.5, 15, and 30 μM doses of nifedipine (Figure 6a).

### 2.10. Nifedipine Upregulated the Expression of SREBP-1 and SREBP-2

The levels of SREBP-1/HDAC and SREBP-2/HDAC were upregulated following nifedipine treatment. SREBP-1 was stimulated in control. However, relative to control, the protein expressions of SREBP-1/HDAC were upregulated to 109.9% and 132.0% at 7.5 and 30 μM of nifedipine, respectively (*p* < 0.01) (Figure 6b); those of SREBP-2/HDAC were elevated to 127.5%, 146.7% and 161.0% of control levels by nifedipine at 7.5, 15, and 30 μM doses, respectively (Figure 6c). The upregulation of both SREBP-1 and SREBP-2 (Figure 6b,c) suggests that both de novo lipogenesis and cholesterol synthesis were stimulated. Consistent with this, the de novo biosynthesis of long chain fatty acid, triglycerides (Figure 4), and cholesterol (Figure 3a and Figure 5) were all stimulated, as evidenced by the upregulated ACC and ACSL1 (Figure 4), as well as ATP citrate lyase and HMG-CoA reductase (Figure 5).

### 2.11. Nifedipine Upregulated the Expression of PPAR-α

Compared with the control group, the level of peroxisome proliferator-activated receptor-α (PPARα)/lamin B was suppressed at a low dose of nifedipine and then dose-dependently restored to 80.2%, 85.6%, and 96.4% upon administration of nifedipine at doses of 7.5, 15, and 30 μM, respectively (*p* < 0.05) (Figure 6d).

### 2.12. Nifedipine Upregulated the Expression of lipin-1

Lipin-1 was activated in control. However, the quantification level of lipin-1/lamin B was upregulated after nifedipine treatment to 102.5%, 115.8%, and 121.1% of control levels by 7.5, 15, and 30 μM of nifedipine, respectively (*p* < 0.05 at both 15 and 30 μM) (Figure 6e).

The summary of nifedipine induced lipogenesis was visually graphed in Figure 7.

## 3. Discussion

In this paper, we described nifedipine, a calcium-channel blocker, as a trigger of lipogenesis and lipotoxicity in rat kidneys. Using normal rat kidney epithelial-derived cell line NRK52E, we demonstrated activation of many enzymes important for lipids’ metabolism. Moreover, higher doses of nifedipine resulted in accumulation of FFA and overproduction of ROS. Simultaneously, p-AMPK activity was inhibited (and, conversely, mTOR was upregulated), which may trigger the transcription of *SREBP-1/2* and *lipin-1*. Inhibition of phosphorylated AMPK activity together with SREBP-1/2, lipin-1 and KIM-1 overexpression are evidence of kidney injury. Overall, in this work, we confirmed the mechanism of action involved in the renal lipotoxicity induced by nifedipine therapy, which can be elicited via the AMPK-SREBP-1/2 pathway.

KIM-1 is not detectable in the normal human and rodent kidney, but its expression increases more than that of any other protein in the injured kidney, and it is localized predominantly to the apical membrane of the surviving proximal epithelial cells [20]. In our experiment, KIM-1 was highly expressed in 48 h compared to 24 h at the baseline, which might have been mediated by high glucose medium (HG) because HG may induce apoptosis and autophagy, and the effect is emphasized by longer duration [21]. There are many reasons why KIM-1 may be released into the circulation after kidney proximal tubule injury. In kidneys with injury, the tubular cell polarity is lost, such that KIM-1 may be released directly into the interstitium. Further, increased transepithelial permeability after tubular injury leads to a backleak of tubular contents into the circulation [22]. Humphreys et al. demonstrated that chronic KIM-1 expression leads to inflammation and tubule interstitial fibrosis, characterized by elevated monocyte chemotactic protein-1 (MCP-1) levels and increased MCP-1-dependent macrophage chemotaxis [23]. In experimental AKI, the intensity of KIM-1 expression increased in proportion to the severity of injury and was consistently present in segment S3 (the collecting tubule, mostly cortical, mainly representing the proximal straight tubule), but only transiently in other segments (i.e., segments S1 and S2, proximal convoluted tubule). Vimentin was absent in the proximal tubules of healthy cats but expressed in injured S3. These findings indicate that S3 is the proximal tubular segment most susceptible to ischemic injury and that KIM-1 is a sensitive tissue indicator of AKI in cats [24]. The upregulation of KIM-1 at 24 h with rapid decline at 48 h may implicate the status of AKI. 

Proximal tubular membrane protein CD36 expression was markedly increased in proteinuric individuals [25]. Hyperlipoproteinemia and its subsequent oxidation are usually associated with glomerular capillary dysfunction and severe glomerulosclerosis in dyslipidemic patients due to lipid deposits in glomeruli [26,27]. Diet-induced hypercholesterolemia led to renal endothelial dysfunction associated with vascular and microvascular remodeling, inflammation, and kidney fibrosis [28]. Palmitic acid and high-fat diets cause lipotoxicity in vivo and in vitro and adversely switch the energy source from the CD36 pathway to the GLUT4 pathway [12]. A recent review by Maréchal et al. emphasizes that growth hormone releasing peptides (GHRP) are a potent inducer of PPAR through activation of CD36 [7], thereby regulating essential aspects of lipid and energy metabolism [29].

Other studies have demonstrated that nifedipine and amlodipine prevent non-esterified fatty acid (NEFA)-induced endothelial dysfunction, leukocyte activation, and enhancement of oxidative stress without affecting blood pressure [30]. As most NEFA are present in form of FFA, we speculate that nifedipine could have inhibited the production of FAS. Thus, cholesterol deposition was increased in the nifedipine-treated group through upregulating the expression of the transcription factor SREBP-2. Of note, the level of FAS was decreased with nifedipine treatment, implying that triglyceride deposits in the kidney were not as high as those in the liver. A similar result was obtained in a study using a high-fat diet-fed animal model, in which kidney triglyceride level was much lower than that of the liver, whereas the opposite trend was observed for cholesterol level [31].

Although prior research demonstrated that nifedipine inhibits oxidative stress via activation of the LKB1-AMPK pathway [32], our findings indicate a lipogenic effect of nifedipine in renal cells. It is known that downregulation of AMPK upregulates mTOR and subsequently stimulates lipogenesis and adipogenesis, resulting in enhanced lipid storage [33]. Maintenance of the energy balance depends on the efficiency of tightly regulated mechanisms of energy intake and expenditure [34]. Establishing the correlation of a high-fat diet with diabetes and the expression of AMPK has been challenging. High-fat diet fed mice demonstrated a marked increase in markers of fibrosis and inflammation, with significantly suppressed AMPK in kidneys [35]. AMPK regulates NFκB activation and acts as a potent regulator of NADPH oxidase as well as the TGF-β system [31]. The activity of AMPK correlates with both obesity-related and diabetes-related renal diseases [31]. Activated AMPK reduced renal hypertrophy. Supposedly, the lipid accumulation in kidneys induced by nifedipine can inhibit the activation of AMPK (Figure 6a), and consequently may provoke early renal damage as caused by a high fat diet [35].

Sterol-regulatory element-binding proteins (SREBPs) are key transcription factors regulating the expression of a diversity of enzymes required for endogenous cholesterol, fatty acid, triglyceride, and phospholipid biosynthesis [36]. SREBP activity is tightly regulated to maintain lipid homeostasis and is modulated by extracellular stimuli such as growth factors. In SREBP processing, the inactive precursors of SREBP transcription factors are synthesized bound to the endoplasmic reticulum (ER) membranes, and their function is mainly controlled by the cellular sterol content. When sterol levels decrease, the precursor is cleaved to activate cholesterogenic genes and maintain cholesterol homeostasis [36]. This sterol-sensitive process appears to be a major point of regulation for the SREBP-1a and SREBP-2 isoforms [36]. SREBP-1a, SREBP-1c and SREBP-2 act differently in lipid synthesis. SREBP-1c is involved in FA synthesis and insulin-induced glucose metabolism (particularly in lipogenesis), whereas SREBP-2 is relatively specific to cholesterol synthesis [37]. The SREBP-1a isoform is associated with both pathways [36]. The unique activation properties of each SREBP isoform facilitate the coordinated regulation of lipid metabolism; however, further studies are needed to understand the detailed molecular pathways that specifically regulate each SREBP isoform [36].

PPARs are ligand-activated transcription factors of the nuclear hormone receptor superfamily comprising the following three subtypes: PPAR-α, PPAR-γ, and PPAR-β/δ [38]. PPARs are involved in various DNA-independent and DNA-dependent molecular and enzymatic pathways in adipose tissue, liver, and skeletal muscle. The PPAR family of nuclear receptors plays a major regulatory role in energy homeostasis and metabolic function [39]. Activation of PPAR-α reduces triglyceride levels and is involved in regulation of energy homeostasis. These pathways are affected in disease conditions and can cause metabolic energy imbalance [39].

Lipins are cytosolic phosphatidate phosphatases (PAPs) involved in the glycerolipid biosynthesis pathway [37]. Proteins in the lipin family play a key role in lipid synthesis due to their PAP activity, and they also act as transcriptional coactivators to regulate the expression of genes involved in lipid metabolism [38]. Lipins in the nucleus act as transcriptional co-activators with peroxisome proliferator-activated receptor γ co-activator-1α (PPAR-γ C-1α (PGC-1α)), PPAR-α, and other factors such as histone acetyltransferase (HAT) to stimulate the expression of genes involved in fatty acid oxidation [19,40]. The lipin proteins usually reside in the cytosol. Translocation of lipin to the endoplasmic reticulum (ER) membrane occurs in response to elevated fatty acid levels. There, it catalyzes the conversion of phosphatidic acid (PA) to diacylglycerol (DAG), building the key substrates for the synthesis of triacylglycerol (TAG), phosphatidylethanolamine (PE), and phosphatidylcholine (PC) [38]. The increased expression of lipin-1 together with the fatty acid-induced translocation of lipin-1 and lipin-2 to the ER facilitates increased TAG synthesis in starvation, diabetes, and stress conditions [40]. 

## 4. Materials and Methods 

### 4.1. Cell Culture

The normal rat kidney epithelial-derived cell line NRK52E (CRL-1571) was obtained from the Bioresource Collection and Research Center, Food Industry Research Development Institute, Hsinchu, Taiwan. NRK52E cells were cultured in 5% bovine calf serum-supplemented Dulbecco’s modified eagle medium (Gibco, Carlsbad, CA, USA) at 37 °C in a humidified atmosphere with 5% CO_2_. Upon reaching 80% confluence, the cells were trypsinized with 0.25% trypsin–0.02% ethylenediaminetetraacetic acid (EDTA) for 5 min at 37 °C and then repassaged.

### 4.2. Cell Survival Assay

NRK52E cells (2 × 10^4^ cells/well) were incubated in a 24-well plate for 16 h, and then treated with the nifedipine (Sigma, St. Louis, MO, USA) at different concentrations (0, 7.5, 15, or 30 μM) or with 150 μM oleic acid (Sigma, St. Louis, MO, USA) as a positive control for 48 h. The nifedipine dose was selected to correspond to internal levels in humans, and the positive control oleic acid dose was selected according to a preliminary 3-(4,5-dimethylthiazol-2-yl)-2-5-diphenyltetrazolium bromide (MTT) test. The cells were washed with phosphate-buffered saline (PBS) and incubated with 0.5 mg/mL MTT solution for 4 h at 37 °C prior to removing the culture medium. Dimethyl sulfoxide was then added and mixed for 5 min at 26 °C. Cell viability was determined by measuring the absorbance at 560 nm. Cell viability for each experimental group was calculated as a percentage of that of the control group.

### 4.3. Oil Red O Staining

NRK52E cells (2 × 10^4^ cells/well) were incubated in a 24-well-plate for 16 h, and then treated with nifedipine (0, 7.5, 15, or 30 μM) or 150 μM oleic acid for 24 and 48 h, respectively. The cells were then fixed on the plate by 10% formalin for 3 h, the plate was washed with 60% isopropanol to dry, and then 300 μL of 3 mg/mL Oil Red O (Sigma) was added to the cells and left for 1 h to air dry. The plate was washed with flowing water and Oil Red O dye, and photographs were acquired. Finally, 200 μL of 100% isopropanol was added to extract residual Oil Red O, and the absorption was monitored at 515 nm.

### 4.4. Lipid Assay

NRK52E cells (2 × 10^4^) were incubated in 24-well plates, and then treated with nifedipine (0, 7.5, 15, or 30 μM) or oleic acid (150 μM) for 48 h. The lipid assay was conducted using the Total Cholesterol and Cholesteryl Ester Colorimetric/Fluorometric Assay Kit from Biovision (Milpitas, CA, USA, #K603-100).

### 4.5. Lipid Peroxidation

The measurement of thiobarbituric acid reactive substances (TBARS) is the most widely employed assay to determine lipid peroxidation by measuring the production of malondialdehyde, which is a naturally occurring byproduct of lipid peroxidation. In this study, we evaluated the extent of lipid peroxidation using a standardized TBARS assay kit (Cayman Chemical, Ann Arbor, MI, USA), according to the manufacturer’s instructions. Cell suspensions were centrifuged at 161× *g* in a fixed angle rotor for 5 min and washed twice with PBS. The supernatants were discarded, and the cell pellets were resuspended in 1 mL of PBS and sonicated on ice. The malondialdehyde thiobarbituric acid adduct formed by the reaction was measured colorimetrically at 530–540 nm.

### 4.6. Measurement of ROS Production

Dihydroethidium (DHE) has been shown to be oxidized by superoxide to form 2-hydroxyethidium (2-OH-E+) (ex 500–530 nm/em 590–620 nm) or by non-specific oxidation by other sources of reactive oxygen species (ROS) to form ethidium (E+) (ex 480 nm/em 576 nm). NRK52E cells (2 × 10^4^) were incubated in 24-well plates and then treated with nifedipine (30 mM) for 16 h (peak ROS level) or with H_2_O_2_ (500 mM) for 30 min as the positive control. The cells were then dyed with the Muse Oxidative Stress Kit (MCH100111; Millipore, Billerica, MA, USA) and subjected to the Muse^®^ Cell Analyzer (Luminex Corp., Austin, TX, USA). All measurements were repeated at least three times. The data were analyzed by the Muse Software module version 1.5.0.0.

### 4.7. Western Blotting Procedure

A standard Western blotting protocol was used as described previously [41,42]. Subcellular fractionation experiments to estimate expression of these molecules in the nuclear fraction including Lipin-1, SREBP-1/2, PPARα using The primary antibodies used in this study included the Fatty Acid and Lipid Metabolism Antibody Sampler Kit (#8335), anti-AMPK (#5832), and anti-p-AMPK (#2535) from Cell Signaling Technology (Danvers, MA, USA); anti-SREBP-1 (sc-13551) and anti-SREBP-2 (sc-13552) (Santa Cruz Biotechnology, Santa Cruz, CA, USA); anti-CD36 (NB400-144ss), Anti-Lipin 1 antibody (ab70138), anti-β actin (NB600-501), and anti-α tubulin (NB100-690, Novus Biologicals, Littleton, CO, USA); anti-PPARα (GTX01098), anti-HDAC1 (GTX100513), anti-lamin B1 antibody (ab65986), and anti-histone H3 (GTX122148) (GeneTex, Irvine, CA, USA); and anti-KIM 1 (ab190696, Abcam, Cambridge, MA, USA). The secondary antibody used was either anti-mouse IgG or anti-rabbit IgG (Jackson ImmunoResearch, West Grove, PA, USA), which was dissolved in 5% skim milk in TBST for 1 h, followed by incubation for 1–2 min in enhanced chemiluminescence mixture (JT96-K004M, T-Pro Biotechnology, Zhonghe, New Taipei City, Taiwan) for visualization. The Western blot was repeated at least three times.

### 4.8. Statistical Analysis

Analysis of variance and post-hoc tests using SPSS 14.0 (SPSS Inc., Chicago, IL, USA) were used to statistically evaluate the data, and results are presented as the mean ± standard deviation. Two-tailed *p*-values ≤ 0.05 (marked as **) were considered statistically significant. In addition, *p*-values ≤ 0.1 (marked as *) are denoted. 

## 5. Limitation

First, in this study, we clarified the lipotoxicity of nifedipine with in vitro cell model. However, to make sure the nifedipine impact on kidney, in vivo experiment must follow. Second, MTT assay is an indirect method to check cell viabilities. Therefore, flow cytometry should be a better approach for further discussion. 

## 6. Conclusions

Lipid accumulation in renal cells has been implicated in the pathogenesis of obesity-related kidney disease; however, nifedipine-induced renal lipotoxicity has never been cited. To our knowledge, this is the first report demonstrating nifedipine-induced lipid accumulation in the kidney. Renal lipotoxicity is a surrogate marker for renal failure or renal fibrosis. Nifedipine induced the accumulation of cytosolic FFA; stimulated the production of ROS, leading to oxidative stress with subsequent cellular cytotoxic lipid aldehydes accumulation; and decreased cell viability and function in kidney [43]. Decreased pAMPK is associated with lipid accumulation and oxidative stress [44].

Prolonged and chronic treatment with nifedipine may induce or potentiate acute kidney injury (AKI), which has been etiologically considered to be closely related to CKD. Our findings suggest possible novel strategies for prevention of such renal lipotoxicity through the administration of certain antioxidant therapies.

## Figures and Tables

**Figure 1 ijms-20-01570-f001:**
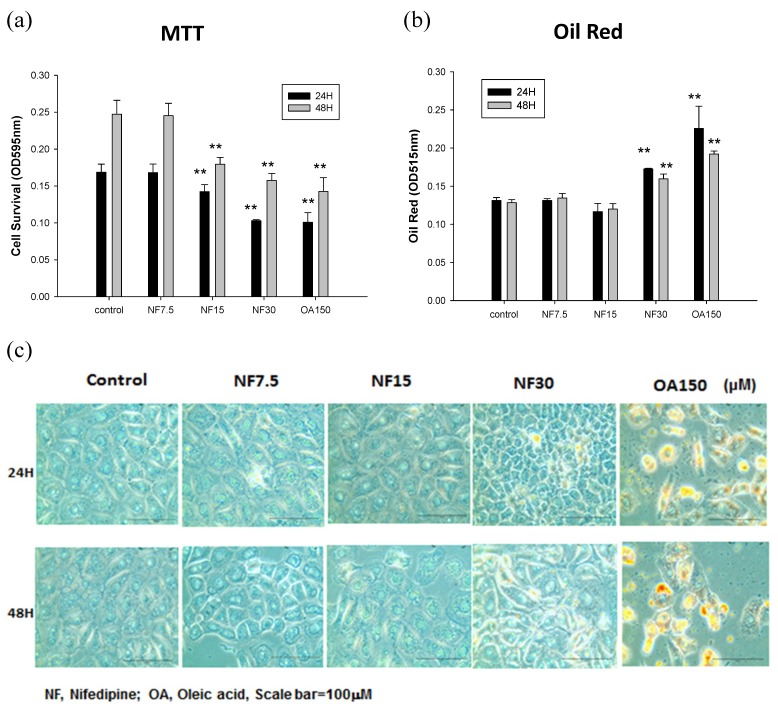
The effect of nifedipine on the cell viability, and intracellular lipid accumulation in NRK52E tubular cells. (**a**) The nifedipine treated cells (15, 30 μM) showed decreased viability compared to control (*p* < 0.01) in 24 or 48 h of MTT assays. (*n* = 3) (**b**) The nifedipine treated cells (30 μM) and oleic acid (150 μM) showed an increasing effect on quantification of Oil Red O staining comparing to control (*p* < 0.01) in 24 or 48 h. (*n* = 3) (**c**) The microscopic Oil Red O staining of NRK52E (magnification, 400×) treated with nifedipine and oleic acid are also shown: (**top**) 24 h; and (**bottom**) 48 h. *p*-values ≤ 0.05 (marked as **) were considered statistically significant. The bar size in (**c**) is 100 μM.

**Figure 2 ijms-20-01570-f002:**
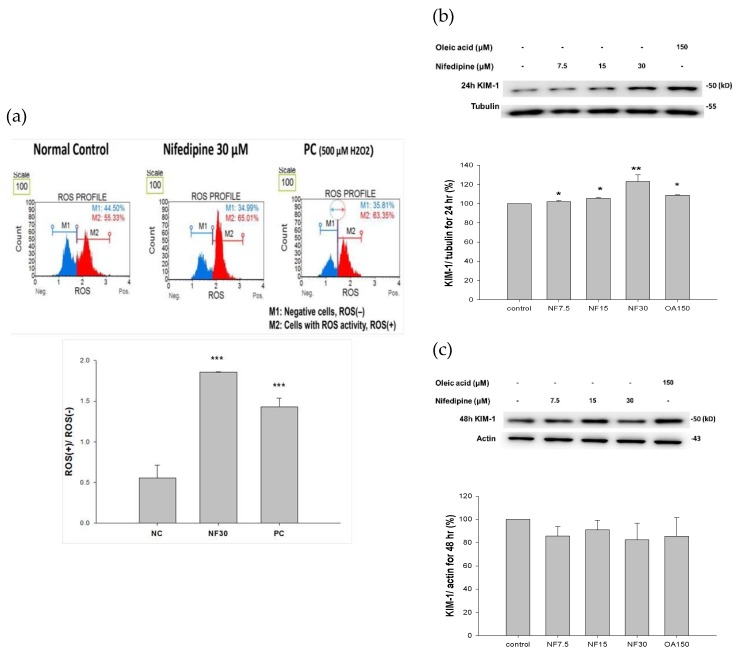
Nifedipine stimulated tremendous production of reactive oxygen species (ROS), and KIM-1 in 24 and 48 h. (**a**) Nifedipine 30 µM-treated group had induced a higher ROS (3.3-fold vs. control, *p* < 0.01) compared to H_2_O_2_ 500 μM. (2.7-fold vs. control, *p* < 0.01). (**b**,**c**) Nifedipine 7.5, 15, and 30 μM-treated groups for 24 h (tubulin as internal control) had upregulated KIM-1 in dose dependent fashion (101%, 102%, *p* < 0.05, and 122%, *p* < 0.01 respectively) and reduced to 86%, 91%, and 80% in 48 h (actin as internal control), respectively. *p*-values ≤ 0.05 (marked as *) were considered statistically significant. In addition, *p*-values ≤ 0.01 are marked as **.

**Figure 3 ijms-20-01570-f003:**
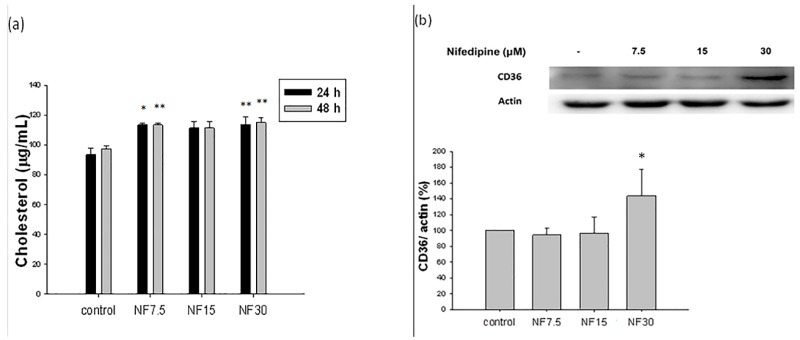
Nifedipine stimulated cholesterol biosynthesis and CD36 protein: (**a**) in a non-dose or time-dependent fashion (111.3 μg/mL in nifedipine 7.5 μM treated group (NF 7.5) and 114.9 μg/mL in nifedipine 30 μM treated group (NF 30), *p* < 0.01 compared to control); and (**b**) CD36 was simultaneously upregulated by nifedipine 30 μM to 143.6% (*p* < 0.05), but unaffected at lower doses ( 7.5 and 15 μM) in 48 h. *p*-values ≤ 0.05 (marked as *) were considered statistically significant. In addition, *p*-values ≤ 0.01 are marked as **.

**Figure 4 ijms-20-01570-f004:**
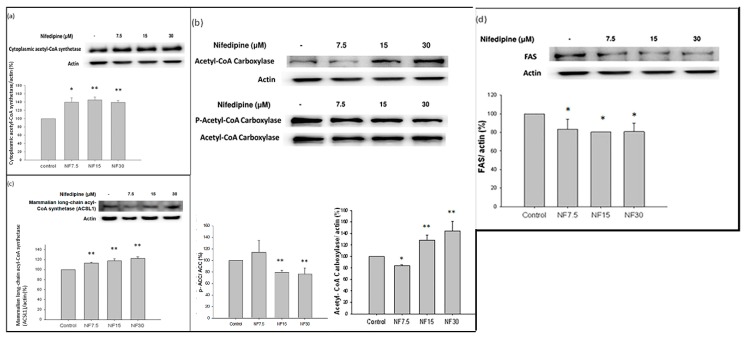
Nifedipine stimulated de novo lipogenesis-1 in 48 h: (**a**) the ACS (*p* < 0.05 by nifedipine 7.5 μM; *p* < 0.01 by 15 and 30 μM); (**b**) the ACC 128.5% and 144.4% at doses 15 and 30 μM, compared to the control (*p* < 0.01), and p-ACC/ACC 81.5% and 74.1% at doses 15 and 30 μM, to the control (*p* < 0.01); (**c**) the ACSL1 showed increased expression of 116.7%, 120%, and 126.7% at doses 7.5, 15, and 30 μM respectively (*p* < 0.01); and (**d**) FAS showed decreased expression of 83.7%, 79.1%, and 81.4% at doses 7.5, 15, and 30 μM respectively (*p* < 0.05). *p*-values ≤ 0.05 (marked as *) were considered statistically significant. In addition, *p*-values ≤ 0.01 are marked as **.

**Figure 5 ijms-20-01570-f005:**
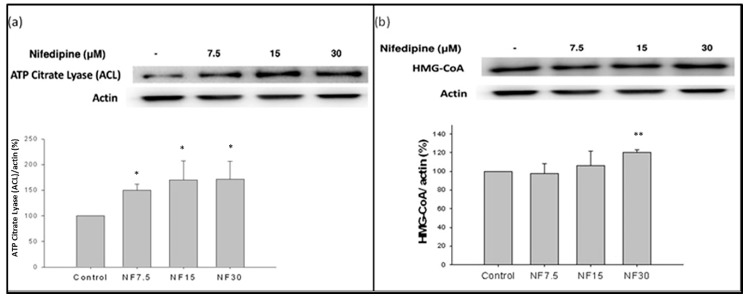
Nifedipine stimulated de novo lipogenesis-2 in 48 h: (**a**) the ACL was stimulated by nifedipine in a dose dependent manner to 150%, 169%, 170% by 7.5, 15, and 30 μM, respectively, compared to the control (*p* < 0.05); and (**b**) the HMG-CoA reductase was also upregulated by nifedipine administration in a dose-dependent fashion up to 107% and 120% at 15 and 30 μM (*p* < 0.05). *p*-values ≤ 0.05 (marked as *) were considered statistically significant. In addition, *p*-values ≤ 0.01 are marked as **.

**Figure 6 ijms-20-01570-f006:**
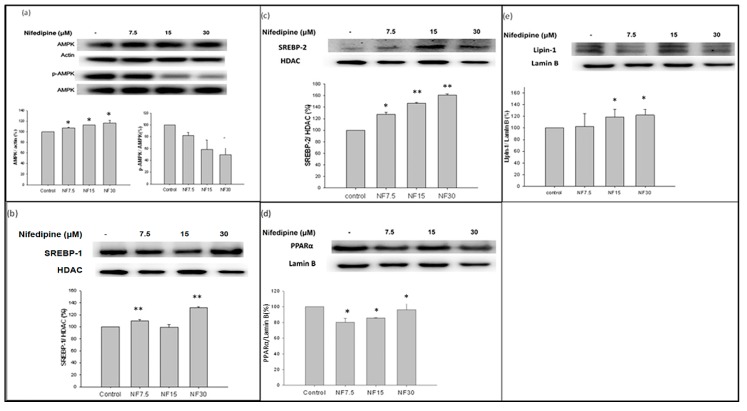
Nifedipine and the expression p-AMPK, SREBP-1/2(nuclear), PPAR-α (nuclear) and lipin-1 (nuclear) in 48 h. (**a**) The AMPK/actin was upregulated to 107.1%, 112.6%, and 116.4%, significantly at doses of 7.5, 15, and 30 μM (*p* < 0.05), the ratio p-AMPK/AMPK was downregulated to 82.1%, 58.7%, and 49.9%, respectively. (**b**,**c**) SREBP-1 was activated in control. The SREBP-1/HDAC was upregulated to 109.9% and 132.0% by nifedipine at doses of 7.5 and 30 μM, respectively (*p* < 0.01); The SREBP-2 were elevated to 127.5%, 146.7% and 161.0% by nifedipine at doses of 7.5, 15 and 30 μM, respectively (*p* < 0.05 in 7.5 μM, *p* < 0.01 in 15 and 30 μM) (**d**) nifedipine downregulated PPAR-α. The PPARα/Lamin B was suppressed from 80.2%, 85.6%, to 96.4% compared to control (100%) upon administration of nifedipine at doses 7.5, 15 and 30 μM, respectively. (*p* < 0.05 in 15 and 30 μM) (**e**) Lipin-1 was activated in control. The lipin-1/lamin B was upregulated after nifedipine treatment to 102.5%, 115.8% and 121.1%, respectively, by nifedipine at doses of 7.5, 15, and 30 μM, respectively (*p* < 0.05 in 15 and 30 μM). *p*-values ≤ 0.05 (marked as *) were considered statistically significant. In addition, *p*-values ≤ 0.01 are marked as **.

**Figure 7 ijms-20-01570-f007:**
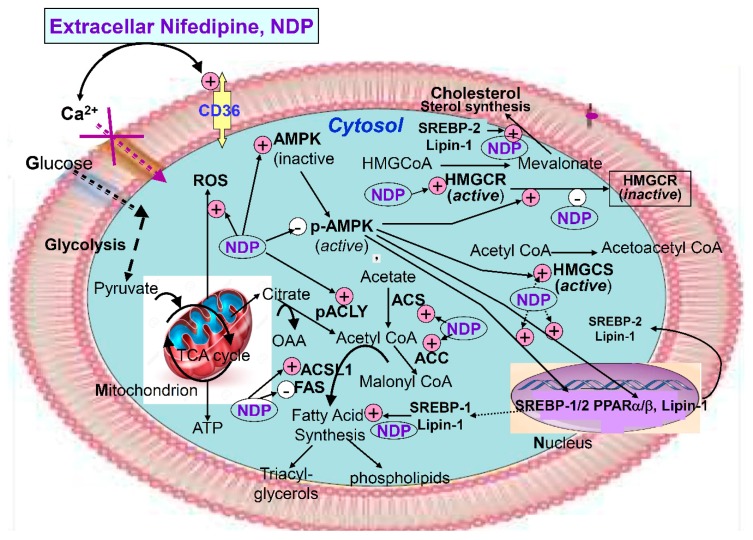
Graphic summary for nifedipine-induced lipogenesis on renal tubular cell. In this study, nifedipine upregulated CD36 membrane protein and ROS, but decreased the active form of p-AMPK. Further, de novo cholesterol biosynthesis by activating nuclear transcriptional factor SREBP-1/2, PPAR-α, and lipin-1 [19] were expressed. Specifically and selectively, SREBP-1a activates fatty acid and cholesterol synthesis; SREBP-1c, fatty acid synthesis; and SREBP-2, cholesterol synthesis and uptake. Then, lipogenesis related enzymes ACS, ACC, ACSL1, ACL, and HMGCR were expressed. NDP, non-dihydropyridine; HMGCR, HMG CoA reductase; ACS, acetyl CoA synthetase; ACC, acetyl CoA carboxylase; ACSL1, long chain fatty acyl elongase; FAS, fatty acid synthase; ACL, ATP citrate lyase; PPAR-α, peroxisome proliferator-activated receptor-α; SREBP1/2, Sterol regulatory element-binding proteins 1/2.

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
