# Peer review of "Nifedipine Modulates Renal Lipogenesis via the AMPK-SREBP Transcriptional Pathway"

_ijms, 2019, doi:10.3390/ijms20071570_

Round 1
Reviewer 1 Report
The manusccript described a surprising finding of an adverse effect of CCB, which stimulates lipogenesis in the kidney to facilitate its damage. The results are provocative and needs its confirmation in vivo studies. Before that, the manuscript can be improved by adding discussion which will provide the limitations of this study as well as the speculative reason for the opposiing results from the previous one. The discussion will be highlighted with Fig. 7 which needs a detailed figure legend including full spells of abbreviation such as NDP (Non-dihydropyridine).
Fig.2 C has a confusion of actin and tubulin.
Author Response
Specific Comment | Response | |
1 | The manuscript described a surprising finding of an adverse effect of CCB, which stimulates lipogenesis in the kidney to facilitate its damage. The results are provocative and needs its confirmation in vivo studies. Before that, the manuscript can be improved by adding discussion which will provide the limitations of this study as well as the speculative reason for the opposiing results from the previous one. The discussion will be highlighted with Fig. 7 which needs a detailed figure legend including full spells of abbreviation such as NDP (Non-dihydropyridine). | We have revised limitation as well as discussions. The full figure legend including full spelling of abbreviation is also provided. Hence, we revised page 12, lines 360-372 and, page 13, lines 421-436, highlighted in Yellow, Thanks for your comment. |
2 | Fig.2 C has a confusion of actin and tubulin. | We have revised figure 2C. Thanks a lot! |
Reviewer 2 Report
Yen-Chung Lin and colleagues described nifedipin, a calcium-channel blocker, as a trigger of lipogenesis and lipotoxicity in rat kidneys. Using normal rat kidney epithelial-derived cell line NRK53E, the authors demonstrated activation of many enzymes important for lipids’ metabolism. Moreover, higher doses of nifedipin resulted in accumulation of FFA and overproduction of ROS. Inhibition of phosphorylated AMPK activity together with SREBP-1/2, lipin-1 and KIM-1 overexpression are an evidence of the kidney injury.
In general, the manuscript of Lin et al. deals with an interesting subject matter, shedding light at the unknown side effect of calcium-channel blockers widely used in hypertension treatment approach. Indeed, it is not too many studies in the knowledge data base (PubMed) showing a link between using of calcium-channel blockers and risk of acute kidney injury, and the present study provides important, new insights into the activation of lipogenesis by nifedipine. However, there are some issues that deserve more attention and that makes feel less enthusiastic about described findings. The methodology used might be significantly improved. Especially, based on the current experimental evidence, I am not convinced that nifedipine treatment causes lipids accumulation in cells. Also, the implication of mitochondria in this process (if any) is not clear.
Major points:
1. While it seems clear that nifedipine affects cells viability in 24h and 48h separately, it is difficult to clarify, if the observed effect is a direct result of nifedipine treatment exactly. Thus, cells viability seems to be significantly higher in the 48h experiment compared to the 24h experiment at the baseline. When compare cells viability in 48h after treatment with different doses of nifedipine or positive control (oleic acid) to 24h untreated control, cells viability is not affected. Please explain the obtained result. To further clarify it, authors should confirm their findings using an alternative technique, for example flow cytometry.
2. Authors speculate on mitochondria dysfunction in the Conclusion section based on ROS production results only. Based on the presented results, authors used DHE staining for flow cytometry to estimate ROS production. Keeping in account, that DHE staining reflects general ROS production, including contribution of ER, it would be better to use Mitosox staining to estimate contribution of mitochondria’s ROS production itself. Additionally, to evaluate mitochondria dysfunction under nifedipine treatment, it would be more informative to perform experiments on estimation of respiratory chain function using Western blot on respiratory complexes CI-CIV and/or estimation of oxygen consumption capability. TEM of mitochondria morphology could also be helpful.
3. When comparing intracellular lipid accumulation in nifedipine treated cells, authors made a conclusion based on Oil Red O staining only. Representative images of stained cells on Fig. 1c do not look consistent with quantification data on Fig. 1b. Thus, nifedipine treatment at dose of 7.5 mM in 48h in the representative image seems to cause more lipid droplets accumulation compared to doses 15 and 30 mM. Moreover, conclusions on cells size and morphology changes based on the Oil Red O staining are not adequate and it would be better to perform phalloidin staining for cells morphology insights or estimate expression of proteins related to cell hypertrophy.
4. In terms of KIM-1 production in NRK53E cells, authors demonstrated significantly increased KIM-1 expression in 48h (Fig. 2c) compared to 24h (Fig. 2b) at the baseline. Please explain observed result. Moreover, presented data reflect in 24h reflected expression KIM-1 with molecular mass around 110 kDa and in 48h around 50 kDa. Does it mean that analyzed cells have difference in the posttranslational modifications of KIM-1 protein in 48h? Please comment.
5. On Fig. 4b, authors demonstrated that expression of total Acetyl-CoA carboxylase is upregulated, while expression of phosphorylated Acetyl-CoA carboxylase is downregulated under nifetidine treatment (however, it is not clear what time point was used, 24h or 48h?). The upper Western blot image indeed demonstrates upregulated expression of total Acetyl-CoA carboxylase, while the bottom Western blot image shows not changes in expression of total Acetyl-CoA carboxylase. This discrepancy should be eliminated. Additionally, quantification analysis of phosphorylated Acetyl-CoA carboxylase over total Acetyl-CoA carboxylase should be presented on the figure. Further, the authors observed dose-dependent activation of long chain fatty acyl elongase under nifedipine treatment (Fig. 4c). I would expect the representative Western blot to be consistent with the quantification.
6. Quantification analysis on Fig. 6b revealed that nifedipine treatment in doses 7.5 and 30 mM enhances expression of SREBP-1, while 15 mM dose has no effect. However, representative Western blot image is not in the consistence with quantification results. What should be done is to generate a more consistent set of blots and repeat the quantification analysis. Similar concern rises in regards of Fig. 6c. Representative Western blot does not reflect quantification data. Thus, on Western blot image 7.5 and 30 mM nifetidine treatments cause lower expression of lipin-1 compared to control, while 15 mM dose causes no changes in lipin-1 expression. Moreover, as you discuss expression of SREBP-1/2, PPARα and lipin-1 expression compared to expression of nuclear markers, it would be more informative to perform subcellular fractionation experiments to estimate expression of these molecules in the nuclear fraction.
7. It would have a lot of benefits if the manuscript language style of the results/discussion description could be revised. In the current version of the manuscript it is difficult to follow authors’ logic and the description of the related literature rather raises more questions than helps to understand obtained results.
8. Figure legend should be revised. Please add brief title sentence for the whole figure and continue with a short statement of what is depicted in the figure, not the results (or data) of the experiment or the methods used. Legend should be detailed enough so that each figure and caption can, as far as possible, to be understood in isolation from the main text.
Minor points:
1. Please explain why did you use 16h time point for nifedipine treatment in ROS production experiments when all other experiments were done using 24 or 48h time points.
2. Page 4, lines 171-176: please split this sentence into few sentences for better understanding of the described results.
3. Page 4, line 182: data on malonyl CoA are not shown, which does not allow you to speculate on it.
4. Page 5, line 218: please correct the font.
5. Page 6, line 284: Oil Red O is not a treatment and using of “enhanced to response to Oil Red” leads to misunderstanding.
6. Page 6, line 285: using word “indicating” is misleading here as you do not show data on TAG or phospholipids levels changes in your experiments.
7. Figure 1c: please indicate the bar size.
8. Figure 2c: Western blot indicates actin as an endogenous control, while quantification analysis is performed over tubulin. Please correct.
9. Figure 3a: please check that units shown for cholesterol levels are correct, as cholesterol levels usually normalized to mg of protein per sample. If it is correct, please explain what mL means in this case (volume of your sample?) and why you used this unit.
10. Figure 3b: please add statistical significance to your NF30 graph.
11. Page 9, lines 323-324: please correct concentration units used for nifedipine treatment.
12. Page 10, line 341: figure legend describes upregulation of PPARg expression, while the image reflects expression of PPARα. Y axis on Fig. 6d is missing its label. Is downregulation of PPAR expression under 7.5 mM treatment is significantly lower compared to control?
13. When describing flow cytometry method, please add information on what flow cytometric analyzer, fluorochrome, laser lenses, emission filters and software for analysis were used.
14. Page 12, line 405: using of mentioned anti-TNFR1 (#13377) antibodies is not reflected in the main text of the paper.
15. For all figure legends, please add information on what statistical test(s) was used. For all figures, please correct the font (some letters and numbers are not readable).
16. Page 13, line 430: please add a reference you cited here.
17. Page 13, line 440: supplementary materials are not provided and the link provided does not work.
Author Response
Yen-Chung Lin and colleagues described nifedipine, a calcium-channel blocker, as a trigger of lipogenesis and lipotoxicity in rat kidneys. Using normal rat kidney epithelial-derived cell line NRK53E, the authors demonstrated activation of many enzymes important for lipids’ metabolism. Moreover, higher doses of nifedipine resulted in accumulation of FFA and overproduction of ROS. Inhibition of phosphorylated AMPK activity together with SREBP-1/2, lipin-1 and KIM-1 overexpression are an evidence of the kidney injury.In general, the manuscript of Lin et al. deals with an interesting subject matter, shedding light at the unknown side effect of calcium-channel blockers widely used in hypertension treatment approach. Indeed, it is not too many studies in the knowledge data base (PubMed) showing a link between using of calcium-channel blockers and risk of acute kidney injury, and the present study provides important, new insights into the activation of lipogenesis by nifedipine. However, there are some issues that deserve more attention and that makes feel less enthusiastic about described findings. The methodology used might be significantly improved. Especially, based on the current experimental evidence, I am not convinced that nifedipine treatment causes lipids accumulation in cells. Also, the implication of mitochondria in this process (if any) is not clear.
Major points:
1. While it seems clear that nifedipine affects cells viability in 24h and 48h separately, it is difficult to clarify, if the observed effect is a direct result of nifedipine treatment exactly. Thus, cells viability seems to be significantly higher in the 48h experiment compared to the 24h experiment at the baseline. When compare cells viability in 48h after treatment with different doses of nifedipine or positive control (oleic acid) to 24h untreated control, cells viability is not affected. Please explain the obtained result. To further clarify it, authors should confirm their findings using an alternative technique, for example flow cytometry. | Thank you for your important suggestions. Initially, higher cell viabilities up to 147.2% in 48 h than in 24 h at control, then nifedipine 30µM significantly decreased cell viabilities (62.5% in 24 h; 60.0% in 48 h) compared to control (p < 0.01). Nifedipine decreased cell viabilities in MTT assays. Hence, we revised page 3, lines 91-98. |
2. Authors speculate on mitochondria dysfunction in the Conclusion section based on ROS production results only. Based on the presented results, authors used DHE staining for flow cytometry to estimate ROS production. Keeping in account, that DHE staining reflects general ROS production, including contribution of ER, it would be better to use Mitosox staining to estimate contribution of mitochondria’s ROS production itself. Additionally, to evaluate mitochondria dysfunction under nifedipine treatment, it would be more informative to perform experiments on estimation of respiratory chain function using Western blot on respiratory complexes CI-CIV and/or estimation of oxygen consumption capability. TEM of mitochondria morphology could also be helpful | Thanks for your great suggestions, we have revised our conclusion that oxidative stress may increase lipid aldehydes through lipid peroxidation and affects cell viability and function [43]. Decreased pAMPK is associated with lipid accumulation and oxidative stress. [44] Hence, we revised page 13, lines 439-432. |
3. When comparing intracellular lipid accumulation in nifedipine treated cells, authors made a conclusion based on Oil Red O staining only. Representative images of stained cells on Fig. 1c do not look consistent with quantification data on Fig. 1b. Thus, nifedipine treatment at dose of 7.5 mM in 48h in the representative image seems to cause more lipid droplets accumulation compared to doses 15 and 30 mM. Moreover, conclusions on cells size and morphology changes based on the Oil Red O staining are not adequate and it would be better to perform phalloidin staining for cells morphology insights or estimate expression of proteins related to cell hypertrophy. | (1) We have checked and renewed representative the Figure 1a. (2) We also revised page 3, lines 100-103. for oil red stain. Thanks for your valuable comments.
|
4. In terms of KIM-1 production in NRK53E cells, authors demonstrated significantly increased KIM-1 expression in 48h (Fig. 2c) compared to 24h (Fig. 2b) at the baseline. Please explain observed result. Moreover, presented data reflect in 24h reflected expression KIM-1 with molecular mass around 110 kDa and in 48h around 50 kDa. Does it mean that analyzed cells have difference in the posttranslational modifications of KIM-1 protein in 48h? Please comment. | (1) KIM-1 is highly expressed in 48h compared to 24h at the baseline, which may be mediated by high glucose medium (HG) because HG may induce apoptosis and autophagy, and the extent will be extended by longer duration [22]. (2) The molecular mass of KIM-1 was 50kDa, we have corrected the wrong typing 110 (KD) in the Figure 2c. Hence, we revised page 11, line 333-336. Thanks for your comments.
|
5. On Fig. 4b, authors demonstrated that expression of total Acetyl-CoA carboxylase is upregulated, while expression of phosphorylated Acetyl-CoA carboxylase is downregulated under nifetidine treatment (however, it is not clear what time point was used, 24h or 48h?). The upper Western blot image indeed demonstrates upregulated expression of total Acetyl-CoA carboxylase, while the bottom Western blot image shows not changes in expression of total Acetyl-CoA carboxylase. This discrepancy should be eliminated. Additionally, quantification analysis of phosphorylated Acetyl-CoA carboxylase over total Acetyl-CoA carboxylase should be presented on the figure. Further, the authors observed dose-dependent activation of long chain fatty acyl elongase under nifedipine treatment (Fig. 4c). I would expect the representative Western blot to be consistent with the quantification. | Thanks for your suggestions. We revised all figure legends by adding 48 h treatment time. We also revised figure 4b by adding quantification analysis of phosphorylated Acetyl-CoA carboxylase over total Acetyl-CoA carboxylase. The ACC increased to 128.5%, 144.4% at doses 15 μM and 30 μM, compared to the control (p< 0.01), and p-ACC/ACC decreased to 81.5%, 74.1% at doses 15 μM and 30 μM, to the control because of inhibitory phosphorylation of ACC (p < 0.01) (Fig 4b). In addition, expressed ACSL1 to 116.7%, 120%, and 126.7% at doses 7.5μM, 15μM, 30μM respectively (p < 0.01 at all concentrations) (Fig 4c). Hence we revised figure 4b and legend, page 4 lines 133-142, and page 7 lines 205-211.
|
6. Quantification analysis on Fig. 6b revealed that nifedipine treatment in doses 7.5 and 30 mM enhances expression of SREBP-1, while 15 mM dose has no effect. However, representative Western blot image is not in the consistence with quantification results. What should be done is to generate a more consistent set of blots and repeat the quantification analysis. Similar concern rises in regards of Fig. 6c. Representative Western blot does not reflect quantification data. Thus, on Western blot image 7.5 and 30 mM nifetidine treatments cause lower expression of lipin-1 compared to control, while 15 mM dose causes no changes in lipin-1 expression. Moreover, as you discuss expression of SREBP-1/2, PPARα and lipin-1 expression compared to expression of nuclear markers, it would be more informative to perform subcellular fractionation experiments to estimate expression of these molecules in the nuclear fraction. | We used subcellular fraction experiments to measure the nuclear fraction of the SREBP-1/2, PPARa and Lipin-1 proteins actually. We also rechecked and renewed SREBP-1 representative figure to get more consistent with the quantification data (Figure 6b). SREBP-1 and lipin-1 were activated in control, and further elevated in 30μM nifedipine in high glucose environment. The WB protein band of lipin-1 in the group of 30μM nifedipine is not obvious due to the bias of internal control of lamin B. However, we did confirm the results of Lipin-1/lamin B by repreated experiments. The protein expression of Lipin-1/lamin B signifantly increased in 15 or 30 μM nifedipine compared to control group.
Hence, we revised figure 6b, and page 4, lines 161-163, page 5, lines 176-178, page 8, lines 220-232, and page 10, lines 301-302. Thanks for your comments.
|
7. It would have a lot of benefits if the manuscript language style of the results/discussion description could be revised. In the current version of the manuscript it is difficult to follow authors’ logic and the description of the related literature rather raises more questions than helps to understand obtained results. | We have rearrangement the result/discussion to make it more readable for our readers. Thank for your recommendations.
|
8. Figure legend should be revised. Please add brief title sentence for the whole figure and continue with a short statement of what is depicted in the figure, not the results (or data) of the experiment or the methods used. Legend should be detailed enough so that each figure and caption can, as far as possible, to be understood in isolation from the main text. | We have revised all captions and figure legends. Thanks for your comments. |
Minor points:
1. Please explain why did you use 16h time point for nifedipine treatment in ROS production experiments when all other experiments were done using 24 or 48h time points. | We used 16h time point for nifedipine treatment in ROS production because it was the time for peak level. Thanks for your comments. |
2. Page 4, lines 171-176: please split this sentence into few sentences for better understanding of the described results. | We have revised the page 4, lines 133-142. Thank you for the great suggestions. |
3. Page 4, line 182: data on malonyl CoA are not shown, which does not allow you to speculate on it. | We have deleted the paragraph describing malonyl-CoA. Thank you for the comments. |
4. Page 5, line 218: please correct the font. | We have corrected the font. Thanks for your comments. |
5. Page 6, line 284: Oil Red O is not a treatment and using of “enhanced to response to Oil Red” leads to misunderstanding. | We have deleted the paragraph using “enhanced to response to oil red”. Thanks for your comments. |
6. Page 6, line 285: using word “indicating” is misleading here as you do not show data on TAG or phospholipids levels changes in your experiments. | We delete the paragraph using “indicating”. Thanks for your comments. |
7. Figure 1c: please indicate the bar size. | The bar size of figure 1c was 100 mm, and we revised the legend of figure 1c. Thanks for your comments.
|
8. Figure 2c: Western blot indicates actin as an endogenous control, while quantification analysis is performed over tubulin. Please correct. | We have revised figure 2c and legends. Thanks for your comments. |
9.Figure 3a: please check that units shown for cholesterol levels are correct, as cholesterol levels usually normalized to mg of protein per sample. If it is correct, please explain what mL means in this case (volume of your sample?) and why you used this unit. | The concentration of cholesterol level use mL because we measured cholesterol level by from uniform 1x106 cells. Thanks for your comments. |
10. Figure 3b: please add statistical significance to your NF30 graph. | We have revised figure 3b. Thanks for your comments. |
11. Page 9, lines 323-324: please correct concentration units used for nifedipine treatment. | We corrected concentration units in the paragraph. Thanks for your comments. |
12. Page 10, line 341: figure legend describes upregulation of PPARg expression, while the image reflects expression of PPARα. Y axis on Fig. 6d is missing its label. Is downregulation of PPAR expression under 7.5 mM treatment is significantly lower compared to control? | We have revised figure 6d Y axis label and legend. Thanks for your comments.
|
13. When describing flow cytometry method, please add information on what flow cytometric analyzer, fluorochrome, laser lenses, emission filters and software for analysis were used. | We have revised the section of Method. “ Dihydroethidium (DHE) has been shown to be oxidized by superoxide to form 2-hydroxyethidium (2-OH-E+) (ex 500-530 nm/em 590-620 nm) or by non-specific oxidation by other sources of reactive oxygen species (ROS) to form ethidium (E+) (ex 480 nm/em 576 nm). NRK52E cells (2 × 104) were incubated in 24-well plates and then treated with nifedipine (30 mM) for 16 h (peak ROS level) or with H2O2 (500 mM) for 30 min as the positive control. The cells were then dyed with the Muse Oxidative Stress Kit (MCH100111; Millipore, Billerica, MA, USA) and subjected to the Muse® Cell Analyzer (Luminex Corp. TX, USA). All measurements were repeated at least three times. The data were analyzed by the Muse Software module version 1.5.0.0.” |
14. Page 12, line 405: using of mentioned anti-TNFR1 (#13377) antibodies is not reflected in the main text of the paper. | We deleted the anti-TNFR1(#13377). Thanks for your comments. |
15. For all figure legends, please add information on what statistical test(s) was used. For all figures, please correct the font (some letters and numbers are not readable). | We have revised all figure legends and fonts to make it more readable. Thanks for your comments. |
16. Page 13, line 430: please add a reference you cited here. | We cited a reference [32] in page 12, lines 370-372. Thanks for your comments. |
17. Page 13, line 440: supplementary materials are not provided and the link provided does not work. | There are no supplementary materials and we deleted the description. Thanks for your comments. |
Round 2
Reviewer 1 Report
The manuscript has been improved.
Reviewer 2 Report
The manuscript by Lin Y.C., et al entitle "Nifedipine modulates renal lipogenesis via the AMPK-SREBP transcriptional pathway" has been revised in accordance with previously suggested points. Description of the background, results and discussion sections is straight and logical and in line with the stated aim of the study. Few minor issues are still raised:
1. Quality of figures should be improved; fonts on some pictures are not readable.
2. Information on Supplementary materials is still appears in the main text. Please correct.